## PERSPECTIVE

# Biomolecular condensates – extant relics or evolving microcompartments?

Vijayaraghavan Rangachari [1✉]

Unprecedented discoveries during the past decade have unearthed the ubiquitous presence of biomolecular condensates (BCs) in diverse organisms and their involvement in a plethora of biological functions. A predominant number of BCs involve coacervation of RNA and proteins that demix from homogenous solutions by a process of phase separation well described by liquid-liquid phase separation (LLPS), which results in a phase with higher concentration and density from the bulk solution. BCs provide a simple and effective means to achieve reversible spatiotemporal control of cellular processes and adaptation to environmental stimuli in an energy-independent manner. The journey into the past of this phenomenon provides clues to the evolutionary origins of life itself. Here I assemble some current and historic discoveries on LLPS to contemplate whether BCs are extant biological hubs or evolving microcompartments. I conclude that BCs in biology could be extant as a phenomenon but are co-evolving as functionally and compositionally complex microcompartments in cells alongside the membrane-bound organelles.

The phenomenon of liquid-liquid phase separation (LLPS) in aqueous solutions has been known for quite some time, especially in the fields of polymer science and process engineering[1–3]. The idea of the demixing of liquids occurring in biology was first postulated by Bugenberg de John and Kurt in 1929[4] and shortly after by Oparin in his book called *The Origin of Life* in 1938[5]. However, it was not until eight decades later in 2009 that unequivocal evidence in support of LLPS and the formation of membraneless organelles (MLOs) in living cells was brought to light by Brangwynne and Hyman[6]. A cornucopia of equally significant discoveries following Brangwynne and Hyman's in the last decade has now unearthed the near-ubiquitous involvement of MLOs in cellular processes across many organisms. Together, these discoveries have not only opened an unprecedented scientific quest to understand the mechanisms of spatiotemporal control in biological processes but also help glean the origins of life itself. MLOs are spatially separated mesoscale compartments devoid of a membrane barrier observed in membrane-bound organelles such as mitochondria or lysosomes. A variety of different cellular processes involve MLOs, which are also referred to as biomolecular condensates (BCs)[7]. In BCs, biomolecules such as proteins and nucleic acids coacervate to demix from the bulk solution by the process that is best described by LLPS (detailed below in the next section). One of the earliest discovered membraneless organelles in eukaryotes is the nucleolus within the nucleus, which is the location of ribosomal synthesis[8]. Since then, more than 20 MLOs have come to the limelight in mammalian cells, and the numbers are rising at a staggering rate[9,10]. The near ubiquity of the phenomenon in a wide range of cellular processes begs the question of whether the coacervation of proteins and RNA toward the formation of BCs and MLOs have come into existence by Darwinian evolution of natural selection for acquiring spatiotemporal control on biochemical processes or are they relics of cellular origins on earth. Given the prevalence and functional integration of this phenomenon in all kingdoms of life, it is important

[1]Department of Chemistry and Biochemistry, School of Mathematics and Natural Sciences and Center for Molecular and Cellular Biosciences, University of Southern Mississippi, Hattiesburg, MS 39402, USA. ✉email: vijay.rangachari@usm.edu

to dwell in some historical, as well as current research perspectives on biomolecular condensates and take a philosophical dive into the possible origins and evolution of BCs. which is likely to kindle intriguing thoughts about this phenomenon in the biological world.

**Physical chemistry of LLPS and condensate formation**. A homogeneous liquid containing a binary mixture of solute and solvent can exist in a fully miscible single phase or undergo LLPS into a demixed, inhomogeneous state containing two or more phases. If the energy of interactions between solute and solvent is greater than that of solute and solute, a single phase will exist but on the other hand, if the energy of solute-solute interactions is greater than that of the solute-solvent interactions, the system will demix and co-exist in two distinct liquid phases[11–13]. For complex multi-component solutions containing two or more biopolymers, the ability of biopolymers containing opposite charges to engage in complex coacervation involving weak, transient, multivalent, and nonstoichiometric interactions comprising of, but not limited to, electrostatic, π-π or cation-π forces, will determine whether they undergo liquid-liquid demixing into two distinct phase regimes[14–16]. LLPS is a density transition with a concentration threshold called saturation concentration ($C_{sat}$) above which the system undergoes LLPS. $C_{sat}$ defines the phase boundary of a particular system[16–18]. Although widely referred to as LLPS, many of the coacervating biomolecules both in vitro and in vivo show viscoelastic properties meaning that the condensates show liquid-like behavior as a function of time and length scales. However, numerous models developed for LLPS adequately describe the macroscopic behavior of the coacervating biomolecules and BCs, and therefore, one may note that within the biological timescales, BCs show liquid-like behavior. Nevertheless, in demixed state, the dense phase is enriched, and the dilute phase is depleted in biopolymers creating a concentration gradient across the phase boundary[19]. The formation of condensates depends on the mixture's composition and other parameters such as temperature and ionic strength. In non-equilibrium states like those in the cells, the phase boundary can be shifted depending on the diffusion and fluxes of molecules and regulators. The demixed state can therefore be reversibly dissolved or formed based on the environmental cues. Demixing of liquids into inhomogeneous two or more component systems by LLPS-like process presents three fundamental advantages in cellular functions: *a*) It offers spatial control via compartmentalization and confinement of biomolecules without having to actively or passively transport them into specialized, membrane-bound organelles[7,10,20], *b*) it enables to achieve an increase in the effective concentrations of coacervating molecules within the dense phase and thereby providing a temporal control over the rates of the reactions[21–23], and *c*) MLOs thus formed as distinct heterogeneous phases can be diffused back to a single homogenous phase by a variety of different mechanisms depending on the environment and stimuli in an ATP-independent manner. These advantages predispose the cellular machinery to adopt phase separation as a cost-effective and need-based mechanism to optimize their functions.

**RNA and pre-biotic world**. It is now well established that RNA molecules are the earliest- biomolecules formed on the planet and were able to sustain primordial life their ability to that catalyze their own self-replication[24–26]. This remarkable discovery of the origins of life on Earth was made by systematically uncovering the complex cellular forms layer by layer to reveal the conserved molecules and machinery across the phylogenic tree. Discoveries such as self-splicing introns in the form of ribozymes[27], and auto-

regulatory riboswitches[28] to name a few, showcase the versatility of RNA molecules. One of the important properties of RNA molecules is the ability self-replicate and a significant body of evidence points toward the evolution of RNA replicase, which is an RNA molecule that is capable of acting as a template for its own replication and the storage of genetic information, and as an RNA polymerase that is able to catalyze its own production[28,29] Although RNAs are versatile molecules that are capable to replicate, catalyze and multiply, these molecules by themselves cannot accomplish more complex functions without achieving some level of selectivity and spatial compartmentalization.

**Evolution of RNA-protein condensates**. One of the most intriguing questions in the transformation of primordial biological reactions occurring in protocells is the evolution of compartmentalization. Despite the self-replicating ability of RNA molecules, an important step toward the evolution of prebiotic molecules to initiate life requires an increase in the effective concentrations of the reactants. It would be necessary for the heterogenous mixture of RNAs and the molecules they interacted with, to confine themselves within microcompartments that can enhance the rate, efficiency, and selectivity. Theories diverge in how protocells evolved in achieving this but based on the evidence one can safely conclude that before the evolution of both lipid-based membrane compartments that furthered into the current day complex eukaryotic cells, simpler non-lipid-based protocells could have evolved to achieve optimization and selectivity for the reactions necessary[30]. If so, what were the methods by which such confinement of molecules especially with RNA was adopted? Evidence indicates that many such membraneless microcompartments could have existed in the prebiotic world in the form of liquid droplets made of organic molecules and oils[31], silica-based inorganic microcells[32,33], aqueous two-phase systems[34], peptide coacervates[32,35], polyester microdroplets from α-hydroxy acids[30], and anhydride compartmentalization mechanisms[34]. Although it will be difficult to track the timeframes of origins for these membraneless microcompartments, parsimoniously one can conclude that they could have coevolved with membrane-bound protocells and compartments. Yet, one of the compelling arguments for the membraneless compartments to have evolved earlier than the membrane-bound ones is the simplicity by which the ions and other molecules can reversibly diffuse in and out of the droplets without spending valuable energy or having to engage active transporters to accomplish sequestration and confinement. In addition, the ease and reversibility of formation and dissolution provide a great deal of temporal and spatial control for the reactions within membraneless microcompartments. One of the pivotal transformations during the compartmentalization period of the prebiotic evolution in gaining spatiotemporal control of biological reactions was the evolution of RNA to protein or the coevolution of RNA-protein complexes, which likely followed the 'RNA-only' world[36–39] (Fig. 1). In either case, the compatibility of RNA to interact with amino acids and peptides was key to organizing into efficient reaction hubs. The highly negatively charged anionic phosphates in monomeric or oligomeric RNA molecules and their 3-dimensional structural plasticity make them highly suitable for electrostatic coacervation with counter-ionic molecules for phase separation, especially cationic amino acids. This complex coacervation phenomenon drives phase separation and the formation of RNA-peptide condensates. During the coevolution of RNA and proteins, evidence indicates that both these molecules interacted extensively in the prebiotic world mainly via such counter-ionic complex coacervation mechanisms[40,41]. In the prebiotic world, cationic depsipeptides, those that contain both

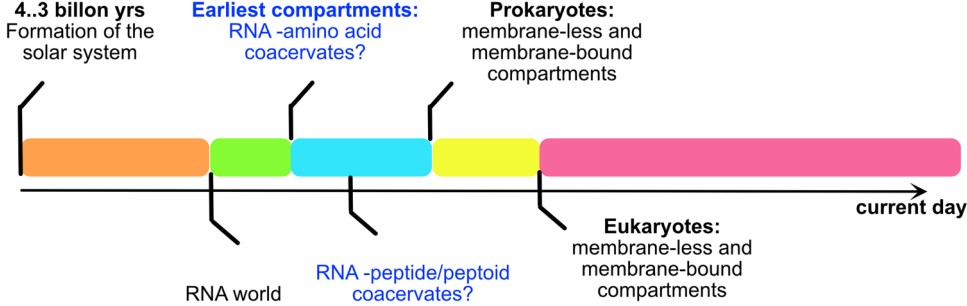

**Fig. 1 Timeline of biomolecular evolution.** RNA-amino acid/RNA-protein condensates during the evolutionary beginnings of life on Earth.

ester and amide linkages formed from hydroxy acids under mild conditions, have been known to be abundant[42–44]. Furthermore, it has recently been identified that cationic amino acids such as lysine, Lys; arginine, Arg; histidine, His; ornithine, Orn; and diamino propionic acid, Dap, interacted with RNA molecules extensively[45]. Among these, Lys and Arg were less likely to be present in the prebiotic world although they have been found in meteorites[46], it is well known that Orn and Dap were abundant and could have formed the earliest RNA-amino acid coacervates[47]. The reminiscence of RNA-peptide condensates' origins is evident when one notices the sheer preponderance of RNA-protein BCs across many organisms[9], which provides unambiguous evidence for RNA-peptide coacervates being one of the earliest microcompartments to be formed in the biological world[48] (Fig. 1).

**The presence of biomolecular condensates across many organisms raises the question of whether they co-evolved with other membrane-bound compartments or are still extant.** BCs and MLOs are prevalent across many organisms in the phylogenetic tree, and discoveries are being made at a staggering rate in many kingdoms of life[49]. In addition to mammalian cells, which I will elaborate on further below, BCs and MLOs are observed in plants[50], bacteria[51], and fungi[52]. Among these, plants are arguably the species that undergo rather unrelenting and constant environmental and climatic stress than others, and therefore, are likely to adopt coping mechanisms that are both reversible and economical. Recent discoveries indicate that plants do contain many different MLOs including, Auxin Response Factor condensates[53], photoreceptor-containing nuclear bodies or "photobodies" that are directly regulated by external light[54], dicing bodies (D-bodies) that are membraneless nuclear hubs for mi-RNA biogenesis in plants[55], in addition to those found in mammalian ones such as Cajal bodies, p-bodies, stress granules, etc.[50] (Fig. 2).

Similarly, advanced single-molecule techniques and super high-resolution microscopy have made it possible to detect MLOs in microorganisms such as bacteria and fungi also. In bacteria, several MLOs, in the form of dense foci, are known to form namely BR-bodies, PopZ microdomains, RNA polymerase clusters, polyphosphate granules, etc. In *Caulobacter crescentus*, BR-bodies are RNA foci assembled by the protein RNase-E that are responsible for mRNA degradation under stress[56]. Also, in *C. crescentus*, MLO-like microdomains formed asymmetrically along the poles by the disordered regions in PopZ, are necessary for the selective localization of many proteins eventually resulting in a skewed inheritance of the transcription factor, CtrA-P in the daughter cells[57,58]. Similarly, RNA polymerases also form liquid-like clusters in *E. Coli* with the help of NusA, a transcriptional termination factor[59]. Furthermore, granules containing inorganic polyphosphates have also been shown to form under cellular stress and starvation to control DNA replication[60]. More recently,

phase separation of the bacterial transcription termination factor Rho in *B. thetaiotaomicron* through its large intrinsically disordered region was found to be key in the survival of the bacteria in the mammalian gut[61]. Similarly, fungi have also been known to contain several MLOs. For example, *S. cerevisiae*, such as the P-bodies, stress granules, nuclear heterochromatin compartmentalization, etc. (reviewed in[52]). The widespread prevalence of BCs across many organisms is probably the consequence of the physiochemical simplicity of the LLPS process which provides a distinctive advantage in sensing and adapting to environmental changes within the cellular milieu without having to expend precious energy. Based on this hypothesis, one could infer that BCs are extant. However, on the other hand, the prevalence of BCs could also suggest their co-evolution along with higher-order organisms containing membrane-bound organelles, as they continue to utilize LLPS and BCs to carry out a multitude of complex functions.

**The spatial and functional ubiquity of biomolecular condensates also prods the question about their evolution.** As detailed above, MLOs are present in many organisms across prokaryotes and eukaryotes and are involved in diverse functions. I will refocus our attention back on mammalian cells to bring out not only the ubiquity involved in the utility of phase separation in cellular functions but also the complexities of the condensates in terms of their composition, size, and biological functions. Although not a prerequisite, intrinsic disorder in proteins facilitates LLPS[62–65]. Based on sequence analysis, it is predicted that disordered proteins are significantly more prevalent in eukaryotes (33%) as compared to prokaryotes (4%)[66], and thus many BCs and MLOs have been identified in mammalian cells.

Among the MLOs in eukaryotes, those that are formed between proteins and RNA are predominant. The most prominent ones discovered so far include Cajal bodies, nuclear speckles, PML bodies and nucleoli in the nucleus, and P-bodies, Balbiani bodies, synaptic densities, RNA transport granules, and germ granules in the cytoplasm[10,67]. In addition to these ubiquitous MLOs, stimuli and condition-dependent ones such as stress granules, U-bodies, metabolic granules, and proteasome storage granules are also formed in various cells[68,69] (Fig. 3). Although many associative polymers and biological molecules can coacervate to form BCs, the ones that are formed between RNA and proteins predominate MLOs in eukaryotic cells[9]. Not only the MLOs are numerous, but they are also equally complex condensates each containing a large number of coacervating biomolecules. The degree of complexity and the ubiquitous nature of BCs can be better appreciated from the following examples which span both physiological and pathological scenarios in eukaryotes.

*Ribosome biogenesis.* Arguably nucleolus is the first MLO to be observed visually. Being the site of ribosome biogenesis, it is now

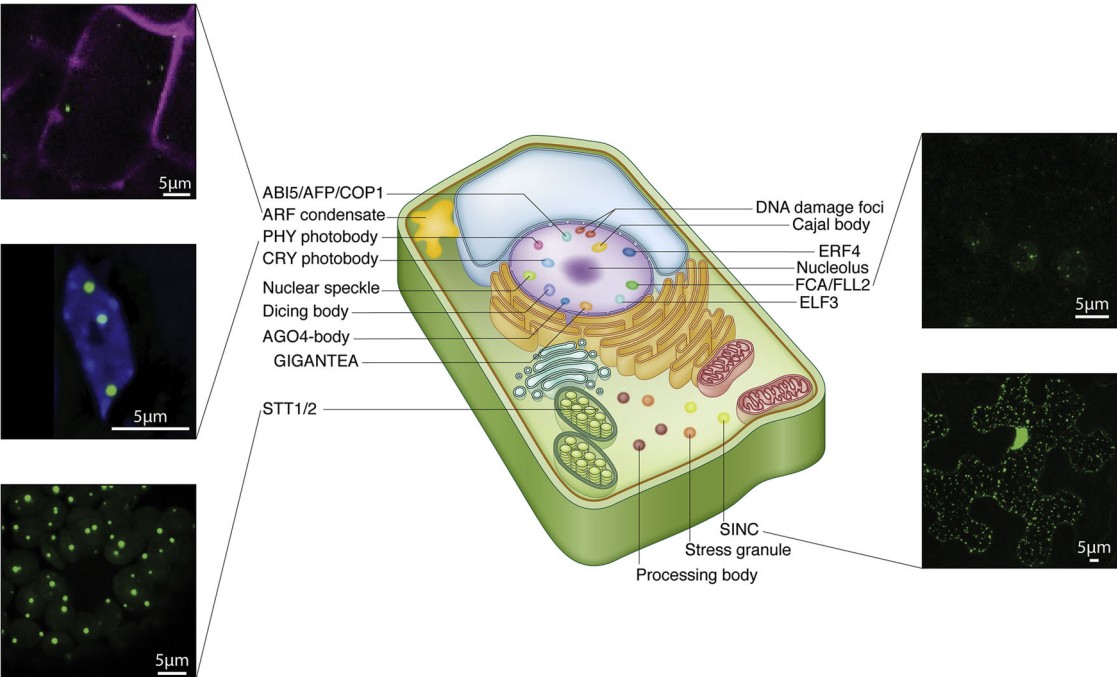

**Fig. 2 MLOs in plant cells.** Simplified cartoon of a plant cell with MLOs depicted along with microscopic images of MLOs of a select few (reproduced with permission from[50]).

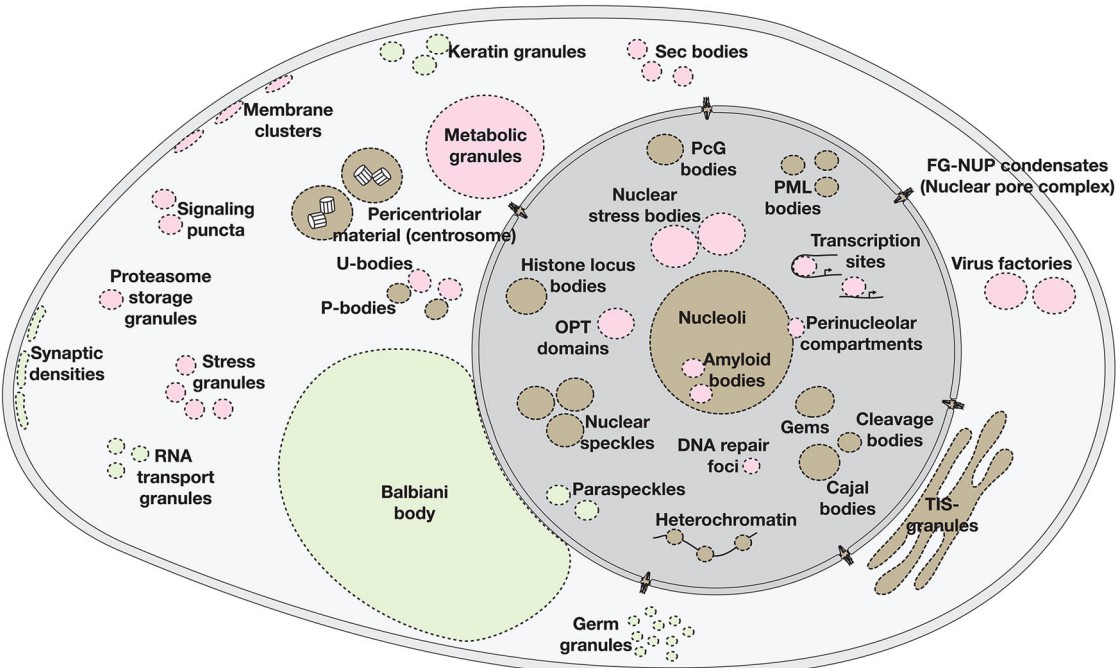

**Fig. 3 MLOs in mammalian cells.** Shown are the condensates that are ubiquitous (brown hues), cell-type specific ones (green hues), and the condition-dependent ones (red hues). (Reproduced with permission from[10]).

clear that nucleolus is also a complex, multi-layer condensate containing three compartments of immiscible liquids with specific events of ribosome biogenesis taking place sequentially from the innermost layer to the outermost layer[8]. The innermost core layer called the fibrillar center (FC) is where the rRNA synthesis begins and as the nascent pre-rRNA emerges, intrinsically disordered Gly-Arg-rich proteins coacervate with them to initiate the formation of the second layer of the nucleolus called the dense fibrillar component (DFC)[70,71].

*Stress response.* These cytoplasmic MLOs are RNA-rich foci that are reversibly formed during cellular stress conditions. The critical role of the stress granules includes the control of translational regulation of mRNA during stress by sequestering the transcripts into the foci[72–74]. They are known to contain more than 25 different proteins, including ribonucleoproteins, those in translationally arrested pre-initiation complex with 40 S ribosomal unit, small RNAs, SG-associated initiation complexes, stress granule nucleators, etc., in addition to mRNA transcripts[72,75].

Proteomic and genetic screens have identified hundreds of proteins associated with stress granules, which require a well-orchestrated play involving physical and chemical forces that synchronize phase separation and one that can adapt to the environmental cues to provide cellular protection under stress. This multi-component, complex biomolecular condensate formation is a key integral part of stress-coping mechanisms in eukaryotic cells.

*Gene regulation.* It has been well-established that transcriptional machinery involved many disordered proteins and disordered regions that provide the plasticity needed for long-range and multivalent interactions between the DNA, RNA, and activation factors[76–78]. Recently, the transcriptional control of transcriptional factors and activating domains embryonic stem cell transcription factor OCT4, estrogen receptor, and yeast transcription factor, GCN4 form phase-separated condensate with other co-activating molecules for gene activation[79]. LLPS-like process has also been observed in key RNA polymerase II transcription[80] and in chromatin regulation[81]. An increasing number of RNA-dependent transcriptional and translational condensate hubs have been discovered[82–84], making the foci containing biomolecular condensates a key component for organizing and activating transcriptional machinery. Related to transcriptional control, the processing bodies or p-bodies are cytoplasmic hubs enriched in mRNA and also play key roles[85,86].

*Catalysis.* A high degree of specificity and selectivity are the key features of enzyme-catalyzed reactions, and therefore, one could assume that enzymes evolved late to carry out such specific reactions. However, since compartmentalization by phase-separated condensates provides higher effective concentrations, enzymes could take advantage of biomolecular condensate as a way of increasing their efficiency. This seems to be the case as many enzymes are known to exist as assemblies and in condensate states[87,88]. Partitioning of enzymes in biomolecular condensates not only facilitates enzyme efficiency by providing increased effective concentrations but also by other mechanisms, such as changing conformations and specific activity[21–23]. An increasing number of reports in addition to these examples showcase the ability of enzymes to be efficient and specific by adopting LLPS-like mechanisms and thus help us glean the versatility of the phenomenon in many cellular aspects. In addition to these, BCs and MLOs are known to play a role in other cellular processes, including autophagy[89], organizing synaptic density[90], and immune response[91].

*Cellular dysfunction and pathology.* One of the obvious consequences of spatial compartmentalization in BCs is the increase in protein concentration in the dense phase. Although this is critical for many physiological processes, BCs become a crucible for many proteins that are prone to form toxic amyloid aggregates[92,93]. Proteins such as FUS[94], TDP43[95], and tau[96] have been shown to form amyloid aggregates nucleating from the BCs and are implicated in pathologies such as amyotrophic lateral sclerosis (ALS), frontotemporal dementia (FTD), and Parkinson's disease (PD)[72]. BCs have also been implicated in cancer[97], and in SARS-Cov-2 viral infection[98].

## Discussion

The afore-described biophysical, spatial, and functional characteristics along with the prevalence of BCs and MLOs in organisms across eukaryotes, prokaryotes, and archaea seem to suggest that the phenomenon of phase separation resembling LLPS is an evolutionarily conserved mechanism for spatial compartmentalization since

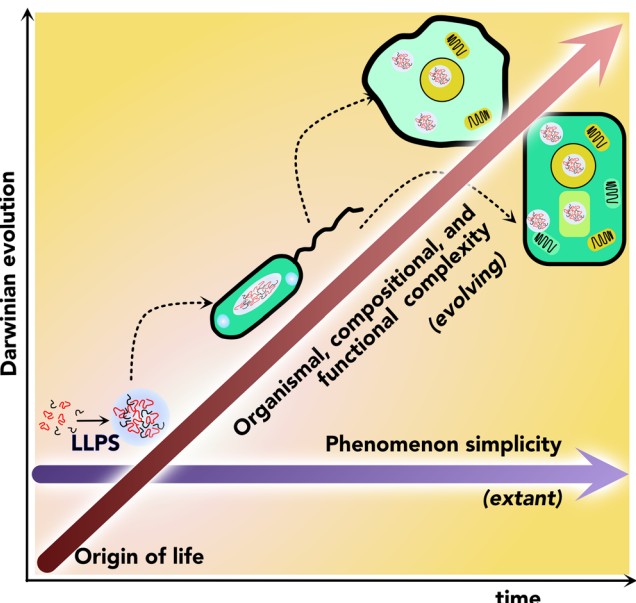

**Fig. 4 Possible evolutionary paths of membranous and membraneless organelles.** The simplicity and ease of spatiotemporal compartmentalization by LLPS remain the key characteristic of BC formation. However, BCs and MLOs have adapted to achieve high levels of compositional and functional complexity in higher-order organisms that seem to co-evolve with membranous organelles. The dotted arrows indicate the longitudinal adaptation of LLPS in organisms.

the primordial life on earth. The most important attribute of phase separation is its simplicity for compartmentalization and spatiotemporal control of coacervating components in a given system as opposed to membrane-bound organelles. LLPS-like process involving simpler two- or three-component systems could have prevailed during pre-biotic and early life processes, but the formation of BCs and MLOs in cellular life forms would have been challenging given the organismal complexity and multi-component systems in a crowded milieu. Yet as described above, widespread BCs are observed in eukaryotes often involved in complex cellular functions and comprising hundreds of components, in some cases containing multiple phases, in a dynamical non-equilibrium system. Ubiquity observed in such complex condensates suggests that cells may have adapted to embrace the tenets of soft matter physics to continue to utilize phenomenologically and energetically simplistic LLPS-like mechanisms rather than the requirement for membrane-bound specialized compartments (Fig. 4). If one were to assume that the phase separation mechanism was out-evolved by the membranous compartmentalization, fewer BCs and MLOs would be observed in eukaryotes and with minimal functional roles, if any. But this does not seem to be the case; not only are BCs prevalent in many cells but also, intriguingly, eukaryotes containing many membrane-bound organelles seem to have integrated phase separation and BCs in their repertoire of biological functions without which the cells may not be able to survive. Some of the complex BCs now observed in many life forms are implicitly dictated and controlled by the sequence and secondary structures of both proteins and RNA –clear evidence for the linked evolution of BCs with the evolution of sequence and structural compositional variance of RNA and protein molecules[99–103]. In addition, the evolution of protein clusters and structural disorder and sequence low complexity also played a role in the evolution of BCs. However, it is also clear that during the early evolution period, LLPS-based compartmentalization also presented limitations on selectivity and specificity, which the membrane-bound organelles could offer. So there seems to be a

trade-off between the energy-efficient, simpler compartmentalization by phase separation and the membrane-bound compartmentalization that offered greater selectivity and specificity but with energy-demanding processes, such as active transport, endo- or exocytosis. Therefore, it seems as if MLOs and membrane-bound organelles have co-evolved compartments instead of out-competing each other. The cells have adapted to utilize both methods of compartmentalization to optimize biological functions depending on the need. Although it would be difficult to assert whether or not BCs are an extant biological relic, or they are evolving alongside the increasing complexities of life forms, it would certainly not be inaccurate to conclude that BCs and MLOs are unlikely to get out-evolved by membrane-bound compartments. It seems as if a synergy has been achieved to utilize both mechanisms to cope and adapt to the increasing complexities of the biological world.

## Data availability

The article contains no new data.

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

## Acknowledgements
The author would like to thank Anukool Bhopatkar, Jhinuk Saha, Shailendra Dhakal, and Faqing Huang for scintillating discussions and inputs.

## Competing interests
The author declares no competing interests.
