## [Peer Review File · Communications Biology]

Reviewers' comments:

Reviewer #1 (Remarks to the Author):

The author has provided a lucid and easily-readable discussion on the possible evolutionary origins of MLOs. The article was easy to read, and raised some interesting ideas.

I have a couple of semantic suggestions, and some additional literature that probably should be cited.

Firstly,

Point 1

The author refers to liquid-liquid phase separation (LLPS) which, I would agree, is the commonly used vernacular. However, whenever people have looked, assemblies formed via phase separation appear to behave as viscoelastic materials where an assignment of solid vs. liquid depends on timescale/lengthscale, as opposed to "simple liquids" that are liquid over essentially any measurable timescale/lengthscale (e.g. water). As such, I might suggest the author makes this clarification or switches from LLPS to simply "phase separation" as a much broader class of processes. Clarification could be obtained simply by saying "While we refer to this process as LLPS in many – if not most – cases, assemblies formed in vitro and in vivo show viscoelasticity, suggesting they are likely not simple liquids, such that these transitions may be better described simply as 'phase separation' without ascribing a specific material state to the two phases. Nevertheless, physical models developed for LLPS have proven highly effective in describing the macroscopic behavior of many BCs, suggesting that on biologically relevant timescales, BCs often appear liquid-like.". Alternatively, the authors could simply refer to phase separation instead of LLPS, although the very nice theoretical description provided by the authors does explicitly describe LLPS. This may seem like semantics (and, in many ways it is), but I think there is some value in this precision and explaining how/why the terms LLPS vs. phase separation are used and why.

Point 2

The unequivocal demonstration that LLPS (or, I might suggest simply 'phase separation') underlies BCs in cells is challenging to obtain, and indeed, BCs do not necessarily need to form via phase separation (as per the original definition by Banani & Lee et al). I absolutely believe phase separation is the most parsimonious explanation, but claims like:

"In BCs, biomolecules such as proteins and nucleic acids coacervate to demix from the bulk solution by the process of LLPS."

Are likely correct but actually lack formal evidence across a wide range of BCs. As such, I might suggest softening this language slightly to something like:

"In BCs, biomolecules such as proteins and nucleic acids coacervate to demix from the bulk solution through a process that is well-described by LLPS."

Again, this is semantics, but protects the author from the criticism of 'reviews saying these bodies are phase separated without any evidence'.

Additional literature

The author may wish to cite/discuss/integrate ideas and discussions from the following relevant papers:

Meyer, M. O., Yamagami, R., Choi, S., Keating, C. D. & Bevilacqua, P. C. RNA folding studies inside peptide-rich droplets reveal roles of modified nucleosides at the origin of life. *bioRxiv* (2023). doi:10.1101/2023.02.27.530264

Cakmak, F. P., Choi, S., Meyer, M. O., Bevilacqua, P. C. & Keating, C. D. Prebiotically-relevant low polyion multivalency can improve functionality of membraneless compartments. *Nat. Commun.* 11, 5949 (2020).

Hansma, H. G. in *Droplets of Life* (ed. Uversky, V. N.) 251–268 (Academic Press, 2023).

Ghosh, B., Bose, R. & Tang, T.-Y. D. Can coacervation unify disparate hypotheses in the origin of cellular life? *Curr. Opin. Colloid Interface Sci.* 52, 101415 (2021).

Antifeeva, I. A., Fonin, A. V., Fefilova, A. S., Stepanenko, O. V., Povarova, O. I., Silonov, S. A., Kuznetsova, I. M., Uversky, V. N. & Turoverov, K. K. Liquid–liquid phase separation as an organizing principle of intracellular space: overview of the evolution of the cell compartmentalization concept. *Cell. Mol. Life Sci.* 79, 251 (2022).

Reviewer #2 (Remarks to the Author):

The work entitled “Biomolecular condensates – extant relics or evolving microcompartments?” by Vijayaraghavan Rangachari provides a comprehensive review of the past and emerging scenario of Biomolecular condensate research. The author infers that biological condensates could be extant as a phenomenon but are co-evolving as 20 functionally and compositionally complex microcompartments in cells alongside the membrane-bound 21 organelles. Overall, the work is presented in a very lucid and exciting way and will likely appeal to a broad group of readerships, which in turn will enable them to think with these new perspectives. The article can be considered for publication but there are certain things which needs to be addressed or I suggest being included.

Some recommended suggestions which can improve the article:

1. Author mentioned “BCs provide a simple and effective means to achieve reversible spatiotemporal control of cellular processes and adaptation to environmental stimuli in an energy-independent manner. The journey into the past of this phenomenon provides clues to the evolutionary origins of life itself.” This section in the abstract is very interesting but the perspective of evolution is introduced in a little abrupt and vague way. It will be a good read if this is done with some perspective (given the word constraints in the abstract) and in the introduction section.
2. Author has stated two broad eras in LLPS/BC research works and has highlighted the work of Brangwyne and Hyman. It is worth discussing the previous works led by the statistical mechanical groups in the introduction which too led to the foundations of the LLPS research.
3. Author has mentioned and reviewed the condensates in the context of evolved cellular systems. It would be interesting if the biomolecular condensates are also discussed in the context of bacterial

systems and how that could likely play an important role in the stress physiology and adaptation in the context of metabolic and physical stress.

4. RNA molecules inherently have their own secondary structures. To interact RNA binding proteins sometimes need to utilize their less structured segment to have conformational complementarity. It would be great if this relation is discussed in some of the segments.

5. Since the article deals with importance of BCs in understanding evolution, it will be relevant to discuss how protein molecules evolved to have differential propensities to form biomolecular clusters in some of the important bio-geological eras of evolution, viz. the great oxidation era, which resulted in the rise of oxygen-based metabolism and an expansion of protein universe.

6. Similarly, it will be interesting to include how BCs in early life forms could have formed multi-functional protein clusters and how that could have impacted evolution.

7. It would be interesting if the sequences promoting condensate formation are scanned for slow codons or kinetic traps and the paralogs are analyzed in the context of kinetics of folding.

8. The article should also highlight how evolutionary co-variation is linked with stretches inside protein molecules which have low complexity regions and function as nucleating sites.

9. Also a section should discuss how protein foldability operates for protein with Low complexity regions and if this is associated with protein evolvability, given low complexity regions can tolerate mutations and can potentially function as mutational capacitors.

Response to the reviewers

I thank the reviewers for their enthusiasm about the perspective and believing that it adds value to the field of biomolecular condensates. I also thank them for their comments and suggestions, which I am addressing here below. The changes are reflected in **red font** in the manuscript.

Reviewer 1:

Point 1: I appreciate the reviewer clarifying and bringing into focus the distinction of LLPS and phase separation of viscoelastic materials. Since most BCs are indeed viscoelastic, I thank the reviewer making us aware of usage of the term LLPS. As per the request, I have not made this clarification on page 3: *“Although widely referred to as LLPS, many of the coacervating biomolecules both in vitro and in vivo show BCs show liquid-like behavior”*.

Point 2: As requested, I have softened our language and connotate with “a process well described by LLPS” and refrained from referring to as LLPS.

Point 3: I have now contextually included the suggested references. Thank you.

Reviewer 2:

Point 1: I thank the reviewer for this suggestion. I have now included a qualifier in the abstract; *“that results in higher concentration and density from the bulk solution”*. Due to word limitations, it could not be expanded further. In the introduction, since I have detailed the process of LLPS in the second section, I have included the pointer, *“(detailed below in the next section)”* right next to the statement on the concept of demixing. I hope this will orient the reader better about the underlying molecular processes on LLPS.

Point 2: Unfortunately, I am not aware of statistical mechanical work on the foundations of LLPS. But if the reviewer could point out specific work, I will be happy to include them.

Points 3: Thank you for the suggestion. I have indeed included some of the important work that has gone into understanding LLPS in bacteria and others such as fungi on page 5. I have now included a few additional and ground-breaking recent work demonstrating how LLPS is an essential mechanism for bacterial to survive in the mammalian gut.

Points 4-9: I appreciate the ideas of expansion proposed by the reviewer and I feel they are certainly inclusion worthy. However, the main objectives of my articles are two folds: a) to succinctly bring out the connection between the BCs that we know now and the coacervates that prevailed during the prebiotic world, and b) to keep it short and relatively unexpansive for readability and engagement. Therefore, I feel that expanding the evolutionary basis further by including sequence biases and mutation tolerance on the low complexity of protein sequences, evolution of protein clusters, codon bias, mutations, oxidative stress, etc., although much relevant, each one is expansive topic in their own merit, and thus dilutes the objective of the perspective. I consciously did not want it to be like a regular review article. But as the reviewer correctly pointed out, inasmuch as the primitive membraneless compartments evolved they did so in conjunction with the evolution of RNA and proteins themselves. Therefore, I have included this important aspect in the discussion with this statements on how RNA and protein structural evolution played a role in BCs: *“Some of the complex BCs now observed in many life forms are implicitly dictated and controlled by the sequence and secondary structures of both proteins and RNA – a clear evidence for the linked evolution of BCs with the evolution of sequence and structural compositional variance of RNA and protein molecules. In addition, evolution of protein clusters and structural disorder and sequence low complexity also played a role in the evolution of BCs”*, and cited several important works in this regard. We sincerely thank the reviewer for pointing out this omission. I hope the reviewer sees the value of keeping the article succinct.

Thank you.

REVIEWERS' COMMENTS:

Reviewer #2 (Remarks to the Author):

I appreciated the changes incorporated in the present version of the manuscript. Also, the author justified as to why certain sections which I recommended for addition is out of scope. The article is likely to be a good read for the community and I recommend its acceptance upon editorial corrections (if any).